# Screening Differential Expression Profiles of Urinary microRNAs in a Gentamycin-Induced Acute Kidney Injury Canine Model

**Bo Sun [1], Liang Chen [1], Zhe Qu [2], Yan-Wei Yang [2], Yu-Fa Miao [2], Rui-Li Wang [1], Xiao-Bing Zhou [2,\*] and Bo Li [2,\*]**

1   College of Bioengineering, Beijing Polytechnic, Beijing 100176, China
2   National Center for Safety Evaluation of Drugs, National Institutes for Food and Drug Control, Beijing 100176, China
\*   Correspondence: zhxb@nifdc.org.cn (X.-B.Z.); libo@nifdc.org.cn (B.L.)

**Abstract:** microRNAs (miRNAs) are promising biomarkers for different pathological models because of their stable and detectable characters in biofluids. Here, we collected urine samples from 5 beagle dogs on the 3th, 6th, and 12th day in an acute kidney injury (AKI) caused by gentamycin. miRNA levels were measured with high-throughput sequencing and the results were then differentially investigated. Gene Ontology (GO) and KEGG pathway analysis were performed to analyze potential target genes corresponding to the differentially expressed miRNAs (DE-miRNAs). Relationships between hub genes and DE-miRNAs were analyzed with STRING and Cytoscape. We identified 234 DE-miRNAs 3, 6, and 12 days after gentamycin treatment ($p < 0.05$). Top 10 up- and down-regulated candidate target genes of DE-miRNAs were predicted by overlapping TargetScan and miRanda results). GO and KEGG analyses for DE-miRNAs demonstrated that the DE-miRNAs target genes are mainly involved in kidney injury-related pathways, such as the insulin signaling pathway, oxytocin signaling pathway, and hedgehog signaling pathway. The network of miRNA-hub genes suggests that miR-452, miR-106a, and 106b participate in regulating the largest number of hub genes. We evaluated the miRNA signature via a canine model built by gentamycin-caused acute kidney injury. Our results represent a valuable resource for evaluating miRNAs as biomarkers of renal toxicity.

**Keywords:** acute kidney injury; high-throughput sequencing; microRNA signature; biomarker

## 1. Introduction

AKI is a common symptom of critical disease and is closely related to mortality. The key feature of AKI is a sudden and sustained decrease in glomerular filtration rate [1]. AKI is common in critically ill COVID-19 patients [2]. Late identification of AKI may contribute to its high mortality, as early stages of the condition may go undetected in traditional diagnostic tests such as serum creatinine concentration. Based on hospitalization data, 19–40% of AKI cases are drug-induced AKI (DIKI) [3]. DIKI is also responsible for a high rate of failures in drug development tests, as it does not respond to several drugs [4]. Research has been focused on exploring an early-onset, sensitive, and specific DIKI biomarker with some markers—such as kidney injury molecule-1 (KIM-1), neutrophil gelatinase-associated lipocalin (NGAL), and albumin—already discovered [5].

microRNAs (miRNAs) are a class of endogenous single-stranded non-coding RNAs (ssncRNAs) that are 18–24 nucleotides long. They play a key role in epigenetic regulation by targeting message RNAs [6]. In recent years, anti-miRNA therapeutic strategies, which require synthesized or modified antisense oligonucleotides, have been on the rise [7]. miRNAs have also been reported as potential biomarkers of human diseases such as head and neck cancer [8], breast cancer, and ovarian cancer [9].

miRNAs including miR-10a, miR-21, and miR-26a have demonstrated anti-inflammatory or anti-apoptotic activities in AKI [10]. The mechanism of vancomycin-induced AKI and tubular cell apoptosis may be related to DNA methylation activated by methyl-CpG binding domain protein 2 (MBD2). Inhibition of MBD2 expression was found to down-regulate the level of miR-301-5p and subsequently restore expression of anti-apoptotic genes [11]. rno-miR-143-3p and rno-miR-122-5p were proposed as potential tubular and rno-miR-3473 as glomerular biomarker candidates in a predominantly tubular (gentamicin and cisplatin) and glomerular (puromycin and doxorubicin) kidney injury study [12]. We have previously studied the expression profile of miRNAs and have identified a number of miRNAs in gentamycin-induced AKI rat [13] and dog [14], which may represent effective biomarkers. In this study, we investigated the expression profile of urinary miRNAs in a canine nephrotoxicity model by using high-throughput sequencing of sRNA. We focused on the potential value of urinary miRNAs as non-invasive biomarkers for nephrotoxicity.

## 2. Materials and Methods

### 2.1. Animals and Ethics Statement

All protocols requiring animals received the approval of the Institutional Animal Care and Use Committee (IACUC) of the National Center for Safety Evaluation of Drugs (NCSED) (IACUC-2019-K003). Twenty healthy male beagle dogs were obtained from Beijing Masi Biotechnology Co., Ltd. (Beijing, China), with the Animal Certificate No. SCXK-(Beijing)2016-0001, and reared as described in previous studies [14]. At the conclusion of the study, all dogs were sacrificed by intravenous sodium pentobarbital.

### 2.2. Study Design

After a 27-day quarantine, the 20 dogs were randomly divided into two groups: a control group ($n = 5$) and an experimental group ($n = 15$). The experimental group was intramuscularly injected with 40 mg/kg gentamycin (Sigma-Aldrich, St. Louis, MO, USA) once a day for 12 days (or until the end of the experiment). The control group was similarly injected with saline daily for 12 days.

Dogs in the experimental group were euthanized on days 3, 6, and 12 ($n = 5$/day) of treatment. All dogs from the control group were euthanized on the 12th day. Serum was collected at specific timepoints and used for biochemical analysis. Serum creatinine (sCr) and blood urea nitrogen (BUN) levels were detected using a Hitachi 7180 automatic biochemical analyzer (Hitachi, Yokohama, Japan). At necropsy, kidneys from all dogs were removed for histopathology. Morphological analysis was performed by a certified veterinary pathologist who was blinded to the animal treatment and biomarker data. Another certified pathologist conducted a blind peer review.

The severity of microscopic changes was evaluated by the study pathologist assigned based on the criteria established by PSTC for defining and reporting grading thresholds. The severity scale ranged from 0 to 5, where Grade 1 (minimal) represents a histopathologic change that is inconspicuous to barely noticeable and occupies approximately 5% of the sectional area. Grade 2 (mild) represents a readily noticeable but not a prominent feature of the tissue and/or may be of no functional consequence, occupying 5–15% of the sectional area. Grade 3 (moderate) represents a prominent but not a dominant feature of the tissue and/or may be considered to have limited impact on organ function, occupying 15–50% of the sectional area. Grade 4 (marked) represents a dominant feature and/or may be considered to cause significant impact on total tissue or organ dysfunction, occupying 50–75% of the sectional area. Finally, Grade 5 (severe) represents a dominant feature and/or may be considered to cause significant impact on total tissue or organ dysfunction, occupying > 75% of the sectional area.

*2.3. RNA Isolation and Sequencing Library Generation*

First, 16-h urine samples were collected between 16:30–18:30 on treatment days 3, 6, and 12 in 50 mL centrifuge tubes on ice. Then, 10–15 mL of the samples was centrifuged for 5 min (4 °C, 650 g), before freezing at −70 °C. The urine supernatant was used as the template for total RNA extraction using an miRNeasy Serum/Plasma Kit (Qiagen, Dusseldorf, Germany). The RNA purity and integrity were detected using 1% agarose gel, and RNA purity was further confirmed using a NanoPhotometer® spectrophotometer (IMPLEN, Westlake Village, CA, USA). The concentration and integrity of RNAs were examined using a Qubit® 2.0 Flurometer (Life Technologies, Carlsbad, CA, USA) and a Bioanalyzer 2100 system (Agilent Technologies, Santa Clara, CA, USA), respectively.

sRNA libraries were constructed using a TruSeq Small RNA Sample Prep Kit (Illumina, San Diego, CA, USA) according to the product manual, and sequenced using an Illumina HiSeq 2000/2500 system (Illumina, San Diego, CA, USA).

The expression levels of miRNA were predicted through TPM (transcript per million). The identification of differentially expressed miRNAs (DE-miRNAs) was carried out through analysis of significant differences ($p < 0.05$) between three groups using the DESeq R package (1.8.3). The $p$ value was corrected using the Benjamini and Hochberg method.

TargetScan (www.targetscan.org, (accessed on 20 February 2021) and miRanda (www.microrna.gr/tarbase, (accessed on 20 February 2021) were used to analyze miRNA–target interactions. The top 10 up- and down-regulated DE-miRNA genes were selected after overlapping TargetScan (default parameters) and miRanda (Max_Energy < −10) analyses.

*2.4. Functional Prediction of miRNA*

Gene Ontology (GO) annotation and Kyoto Encyclopedia of Genes and Genomes (KEGG) pathway analyses were used to predict roles of 20 DE-miRNAs selected using the DAVID 6.8 database for visualization, integrated discovery, and annotation.

*2.5. Protein–Protein Interaction (PPI) and miRNA Regulatory Networks Analysis*

The STRING database was used to build the PPI network of the relationship among op 10 up- and down-regulated DE-miRNAs [15]. In this analysis, the connectivity degree of hub genes as well as construction of interactive networks were performed by Cytoscape software (V3.7.1, National Institute of General Medical Sciences, Boston, MA, USA) based on the screened DE-miRNAs and the top 10 hub genes.

*2.6. Statistical Analysis*

The miRNA profile was obtained on day 6 and day 12, using day 3 as the baseline. Transcripts per million (TPM) measurement was used to estimate a given miRNA's expression level. A heatmap of DE-miRNAs was drawn using the ID of DE-miRNA as the abscissa and the groups as the ordinate. One-way ANOVA (unpaired) was used to compare the difference in miRNA expression between groups. The selection of top 10 up- and down-regulated DE-miRNAs was based on 2-fold change (linear) and ANOVA $p < 0.05$.

## 3. Results

### 3.1. Pathological Feature

Kidney tissues were collected from control and experimental groups on 3, 6, and 12 days after treatment with gentamycin. The results of hematoxylin and eosin (H&E) staining are shown in Figure 1. It was found that pathological features were unchanged on day 12 in the control group (Figure 1A). However, the proximal convoluted tubule of dogs in the experimental group had changes in tubular cell degeneration/necrosis, and tubular cells present at days 6 and 12 (Figure 1C,D). From the histopathological scores, minimal renal injury occurred on day 6 and mild injury on day 12 (Table 1).

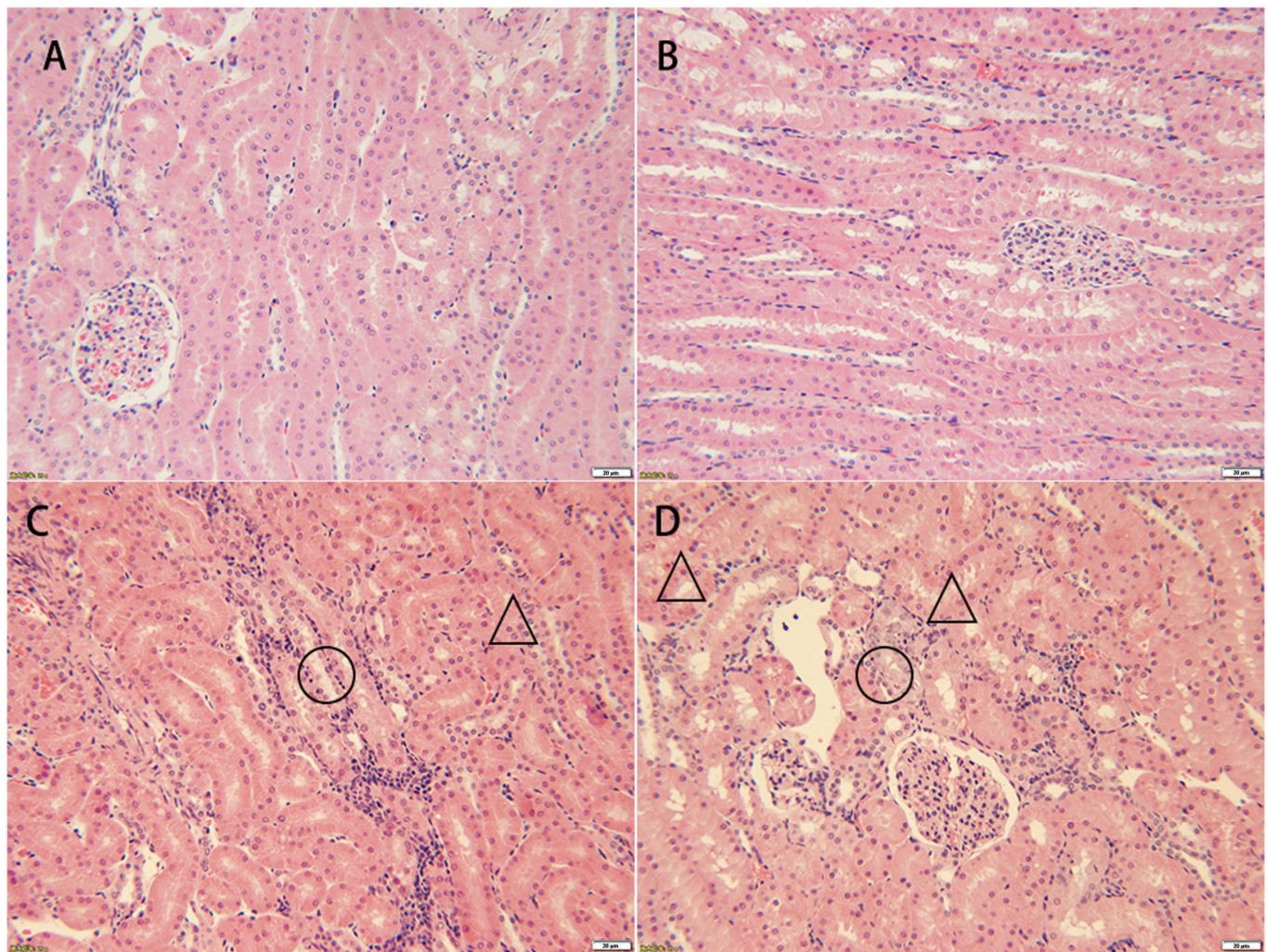

**Figure 1.** Pathological examination of kidney tissues from beagle dogs treated with gentamycin (40 mg/kg/day). No obvious changes are observed in the control group (**A**) and the experimental group on day 3 (**B**). Tubular cells necrosis (circle) and hyaline droplet (triangle) are revealed in the experimental group on day 6 (**C**) and day 12 (**D**). HE 200×, each bar indicates 20 μm.

**Table 1.** Histopathological scores of kidney in beagle dogs treated with gentamycin (40 mg/kg/day).

| Histopathological Change | Control | Experimental | | |
|---|---|---|---|---|
| | | **Day 3** | **Day 6** | **Day 12** |
| Tubular cell degeneration/necrosis | no lesion (5/5) | no lesion (5/5) | no lesion (1/5) | minimal (1/5) |
| | | | minimal (4/5) | mild (4/5) |
| Tubular cell hyaline droplet | no lesion (5/5) | no lesion (5/5) | no lesion (3/5) | |
| | | | minimal (2/5) | mild (5/5) |

### 3.2. Biochemistry Profile

There were no significant differences in expression levels of routine serum indexes (sCr and BUN) between the two groups at each time point ($p > 0.05$) (Figure 2).

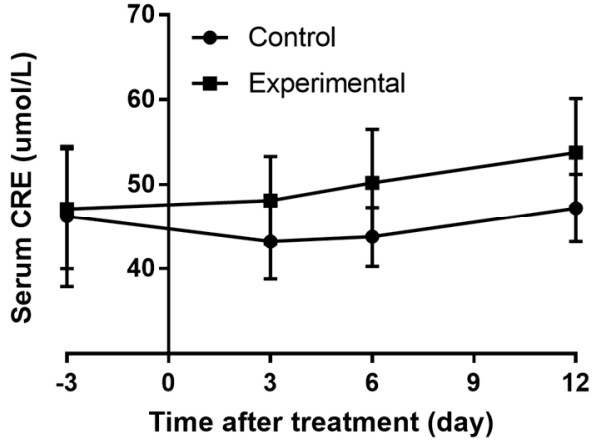 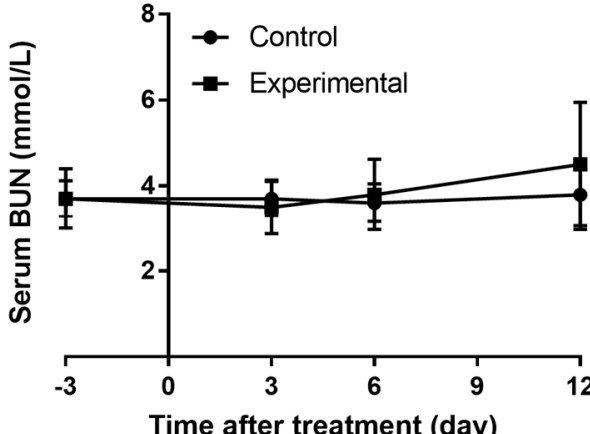

**Figure 2.** The trend of serum CRE and BUN levels in beagle dogs treated with gentamycin (40 mg/kg/day).

### 3.3. miRNA Expression Profile

Urine samples from beagle dogs were collected on days 3, 6, and 12 after the first injection of gentamycin to identify potential urinary miRNAs with altered expression levels related to kidney injury. Clustering analysis is visualized in a heatmap shown in Figure 3. The top 10 up- and down-regulated DE-miRNAs were selected and shown in Table 2.

**Table 2.** The top 10 up- and down-regulated DE-miRNAs.

| No. | miR-ID | Expression | Log2 Fold Change | | *p*-Value |
|---|---|---|---|---|---|
| | | | Day 6 vs. Day 3 | Day 12 vs. Day 6 | |
| 1 | cfa-miR-30a-3p | UP | 2.06 | 0.39 | $3.06 \times 10^{-4}$ |
| 2 | cfa-miR-200a-3p | UP | 1.51 | 0.50 | $3.82 \times 10^{-4}$ |
| 3 | cfa-miR-192 | Up | 1.26 | 0.43 | $3.50 \times 10^{-3}$ |
| 4 | cfa-miR-10b | Up | 3.37 | 0.10 | $6.72 \times 10^{-3}$ |
| 5 | cfa-miR-378 | Up | 2.00 | 1.48 | $7.55 \times 10^{-3}$ |
| 6 | cfa-miR-194 | Up | 2.18 | 0.08 | $8.76 \times 10^{-3}$ |
| 7 | cfa-miR-30a | UP | 0.75 | 0.46 | $9.85 \times 10^{-3}$ |
| 8 | cfa-miR-200c | UP | 1.12 | 1.20 | $1.36 \times 10^{-2}$ |
| 9 | cfa-miR-30c-2-3p | Up | 2.08 | 0.08 | $2.10 \times 10^{-2}$ |
| 10 | cfa-miR-452 | UP | 2.02 | 0.29 | $2.63 \times 10^{-2}$ |
| 11 | cfa-miR-106b | Down | −2.08 | −0.02 | $1.53 \times 10^{-6}$ |
| 12 | cfa-miR-16 | Down | −1.52 | −0.03 | $7.49 \times 10^{-6}$ |
| 13 | cfa-miR-15b-3p | Down | −1.63 | −0.45 | $5.69 \times 10^{-5}$ |
| 14 | cfa-miR-15b | Down | −1.38 | −0.05 | $1.68 \times 10^{-4}$ |
| 15 | cfa-miR-18a | Down | −1.54 | −0.76 | $1.77 \times 10^{-4}$ |
| 16 | cfa-miR-106a | Down | −1.39 | −0.44 | $1.28 \times 10^{-3}$ |
| 17 | cfa-miR-142 | Down | −1.18 | −0.06 | $1.38 \times 10^{-3}$ |
| 18 | cfa-miR-26a | Down | −1.04 | −0.99 | $1.22 \times 10^{-2}$ |
| 19 | cfa-miR-424-5p | Down | −1.46 | −0.24 | $3.81 \times 10^{-2}$ |
| 20 | cfa-miR-101 | Down | −0.57 | −0.40 | $3.06 \times 10^{-4}$ |

### 3.4. Functional Prediction of miRNAs in Gentamycin-Induced AKI

Target genes identified by TargetScan and miRanda were overlapped; the top 10 up- and down-regulated DE-miRNAs were considered. We performed GO functional annotation and KEGG pathway enrichment analyses using DAVID. The results are shown in Figure 4.

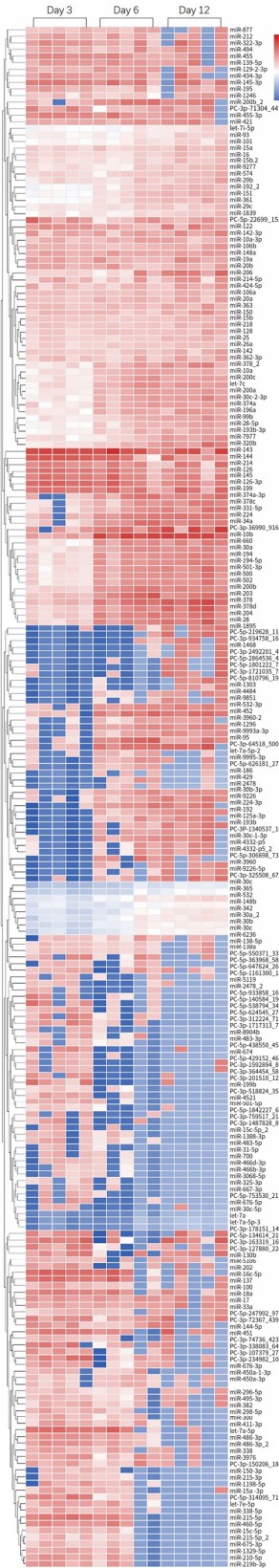

**Figure 3.** Heatmap of DE-miRNAs on the 3th, 6th, and 12th day from AKI which was induced by gentamycin. Each column refers a sample and each row represents a miRNA. 'Red' and 'blue' indicate high and low relative expression, respectively.

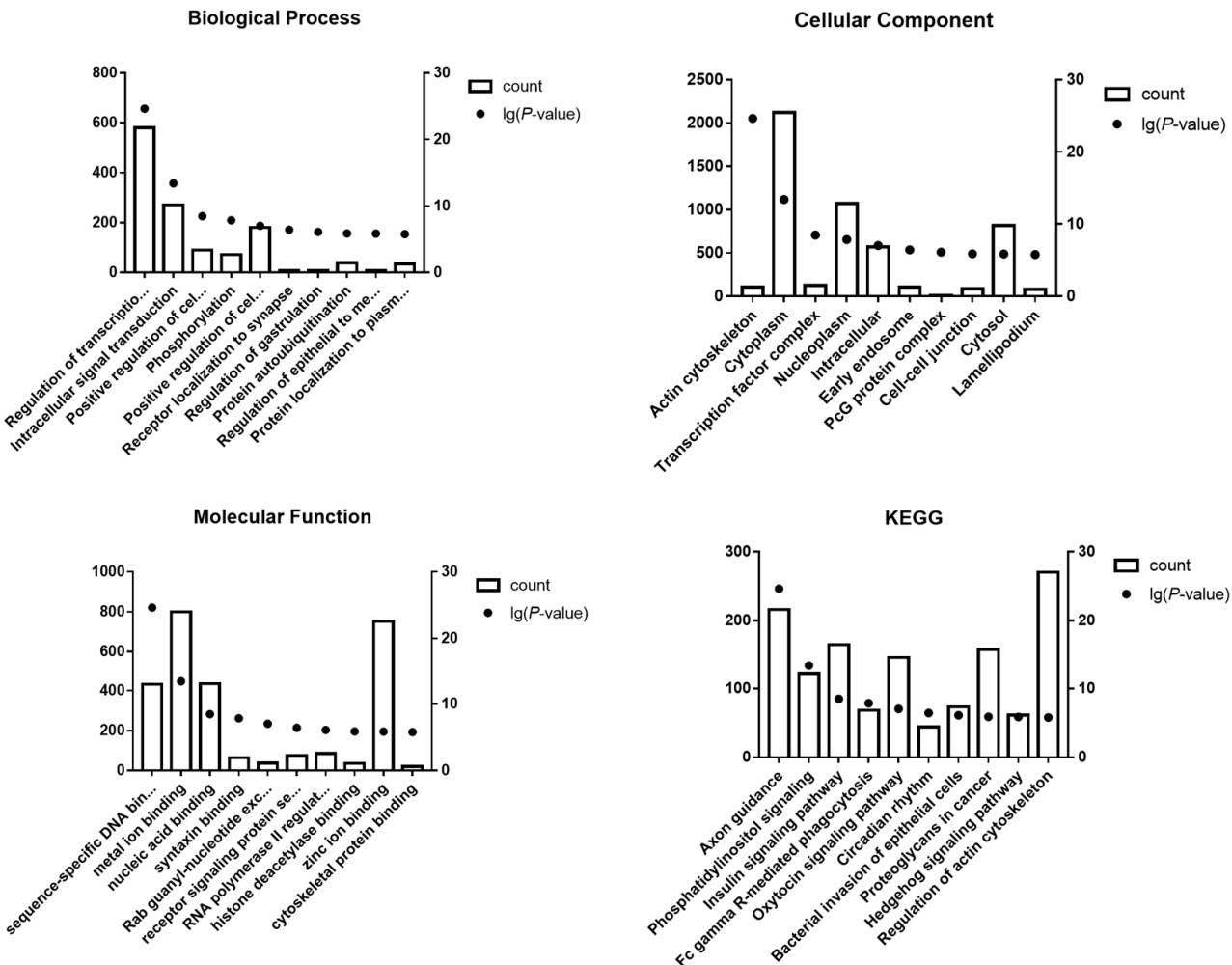

**Figure 4.** GO annotation and KEGG pathway enrichment analysis for the target genes of the top 10 up-regulated and down-regulated DE-miRNAs.

We selected three GO categories from the GO functional annotation, including biological process (BP), cellular component (CC), and molecular function (MF). GO analysis is depicted in Figure 4 and summarized in Table 3.

BP analysis demonstrated regulation of various processes including transcription, intracellular signal transduction, positive regulation of cell proliferation, phosphorylation, cell migration, receptor localization to synapse, regulation of gastrulation, protein autoubiquitination, regulation of epithelial to mesenchymal transition, and protein localization to plasma membrane. CC showed involvement of various cellular components such as the actin cytoskeleton, cytoplasm, transcription factor binding, nucleoplasm, intracellular, early endosome, PcG complex, cell–cell junction, cytosol, and lamellipodium. MF analysis revealed activities such as sequence-specific DNA binding transcription factor activity, metal ion binding, nucleic acid binding, syntaxin binding, Rab guanyl nucleotide exchange factor activity, receptor signaling protein serine/threonine kinase activity, RNA polymerase II regulatory region sequence-specific DNA binding, histone deacetylase binding, zinc ion binding, and cytoskeletal protein binding.

KEGG pathway analysis identified 10 enrichment pathways including axon guidance, phosphatidylinositol signaling pathway, insulin signaling pathway, Fc gamma R-mediated phagocytosis, xytocin signaling pathway, circadian rhythm, bacterial invasion of epithelial cells, proteoglycans in cancer, hedgehog signaling pathway, and regulation of actin cytoskeleton, as depicted in Figure 4 and summarized in Table 4.

**Table 3.** GO functional annotation.

| GO Categories | GO Term | GO ID |
|---|---|---|
| Biological Process | Regulation of transcription | 0006355 |
| | Intracellular signal transduction | 0035556 |
| | Positive regulation of cell proliferation | 0008284 |
| | Phosphorylation | 0016310 |
| | Positive regulation of cell migration | 0030335 |
| | Receptor localization to synapse | 0097120 |
| | Regulation of gastrulation | 0010470 |
| | Protein autoubiquitination | 0051865 |
| | Regulation of epithelial to mesenchymal transition | 0010717 |
| | Protein localization to plasma membrane | 0072659 |
| Cellular Components | Actin cytoskeleton | 0015629 |
| | Cytoplasm | 0005737 |
| | Transcription factor binding | 0008134 |
| | Nucleoplasm | 0005654 |
| | Intracellular | 0005622 |
| | Early endosome | 0005769 |
| | PcG protein complex | 0031519 |
| | Cell–cell junction | 0005911 |
| | Cytosol | 0005829 |
| | Lamellipodium | 0030027 |
| Molecular Functions | Sequence-specific DNA binding transcription factor activity | 0003700 |
| | Metal ion binding | 0046872 |
| | Nucleic acid binding | 0003676 |
| | Syntaxin binding | 0019905 |
| | Rab guanyl-nucleotide exchange factor activity | 0017112 |
| | Receptor signaling protein serine/threonine kinase activity | 0004702 |
| | RNA polymerase II regulatory region sequence-specific DNA binding | 0000977 |
| | Histone deacetylase binding | 0042826 |
| | Zinc ion binding | 0008270 |
| | Cytoskeletal protein binding | 0008092 |

**Table 4.** KEGG pathway.

| KEGG Pathway | KEGG ID |
|---|---|
| Axon guidance | ko04360 |
| Phosphatidylinositol signaling pathway | ko04360 |
| Insulin signaling pathway | ko04910 |
| Fc gamma R-mediated phagocytosis | ko04666 |
| Oxytocin signaling pathway | ko04921 |
| Circadian rhythm | ko04710 |
| Bacterial invasion of epithelial cells | ko05100 |
| Proteoglycans in cancer | ko05205 |
| Hedgehog signaling pathway | ko04340 |
| Regulation of actin cytoskeleton | ko04810 |

*3.5. Regulatory Network of miRNAs in Gentamycin-Induced AKI*

The top 10 hub nodes were screened using Cytoscape software. These data are shown in Table 5. *GATA4* had the highest node degree (58) across the top 10 hub genes identified as targets of up-regulated miRNAs, while *MAPK1* had the highest node degree (336) among target genes of down-regulated miRNAs.

Networks of miRNA-hub genes were constructed to visualize interactions, as shown in Figure 5. Among the top 10 up-regulated miRNAs, cfa-miR-452 and cfa-miR-30a regulated the most hub genes (*n* = 5). cfa-miR-106a and cfa-miR-106b were involved in expression regulation of the most hub-genes among down-regulated miRNAs (*n* = 5).

**Table 5.** The top 10 hub genes identified in the PPI networks.

| Up-Regulated miRNAs | | Down-Regulated miRNAs | |
|---|---|---|---|
| **Gene Symbol** | **Degree** | **Gene Symbol** | **Degree** |
| GATA4 | 58 | MAPK1 | 336 |
| GATA3 | 53 | PHLPP2 | 277 |
| TRPS1 | 51 | EHMT2 | 239 |
| CAPN7 | 48 | RAC1 | 229 |
| IGF1R | 44 | RHOC | 201 |
| LYN | 37 | ACTL7B | 197 |
| RPS6KA2 | 35 | GSK3B | 192 |
| ERBB4 | 34 | PIKFYVE | 186 |
| RPS6KA3 | 33 | KRAS | 185 |
| RAP2C | 32 | FYN | 179 |

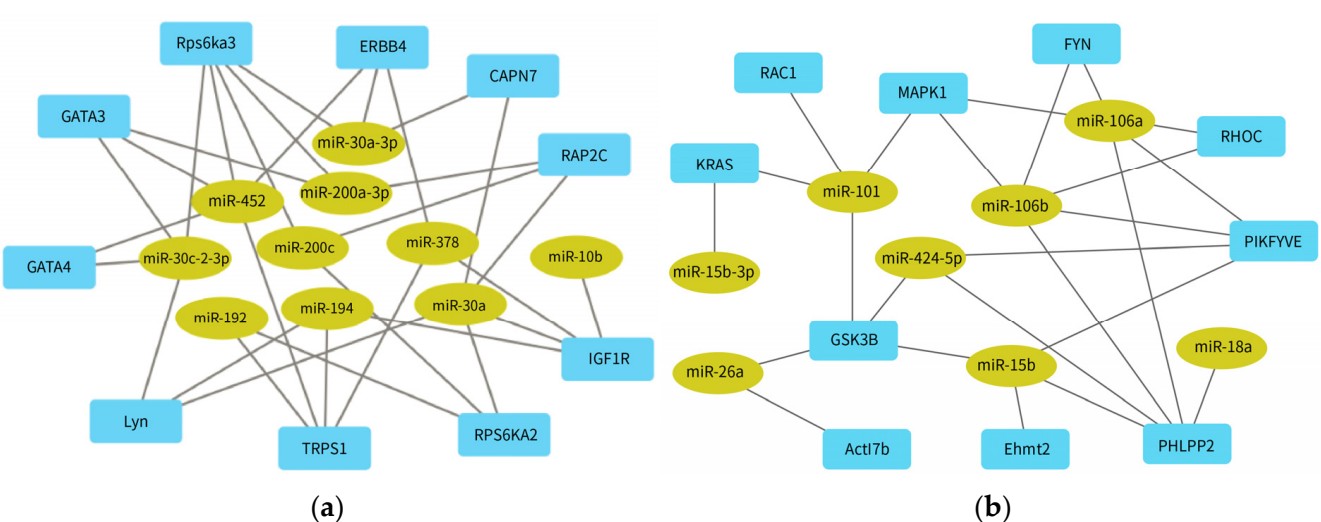

(**a**)  (**b**)

**Figure 5.** miRNA-hub genes network for (**a**) the top 10 up-regulated DE-miRNAs and (**b**) the top down-regulated DE-miRNAs.

## 4. Discussion

One of the primary functions of the kidneys is to concentrate and eliminate various metabolites, including drugs. The clinical risk factors for AKI mostly focus on sepsis, ischemia-reperfusion (I/R), and different drugs. A study conducted on critically ill pediatric patients in China indicated that drugs such as furosemide, midazolam, and fentanyl citrate were more likely to induce DIKI [16]. Although there are numerous drugs that can potentially induce nephrotoxicity in clinical practice, the mechanism of kidney injury is relatively straightforward. Gentamycin, the drug used in our study, is mainly absorbed in the proximal tubules, which can cause tubular cell toxicity and eventually induce AKI [17]. Based on the pathological examination results, intravenous injection of 40 mg/kg gentamicin over 12 consecutive days can be used to recreate the DIKI model in Beagle dogs (Figure 1), and the degree of injury ranged from minimal to mild (Table 1). Although the levels of sCr and BUN increased on day 6 and day 12 after treatment with gentamycin, there was no statistically significant difference between the two groups (Figure 2). Traditional serum biomarkers are not reliable in accurately diagnosing AKI, especially in the early stages of gentamycin-induced nephrotoxicity. Several novel non-invasive bio-fluid biomarkers including cystatin C (CysC), metalloproteinase-2 (TMP-2), insulin-like growth factor binding protein 7 (IGFBP7), KIM-1, and NGAL have been validated for their potential applications in diagnosing DIKI [18].

miRNAs fulfill the FDA standards for ideal biomarkers, including characteristics such as accessibility in various body fluids, stability, cross-species conservation, association with specific tissues or pathological states, and availability of sensitive measurement methods [19]. In addition, different levels of circulating miRNAs can predict tissue-specific pathological changes [20]. Various studies have identified circulating miRNAs as potential biomarkers for different pathological conditions, such as cancers [21], neurodegenerative diseases [22], and hematological diseases [23].

The miRNA expression profile of an AKI canine model induced by gentamycin was examined by RNA sequencing and identified the top 10 DE-miRNAs that were most up- and down-regulated (Table 1). It is noteworthy that miR-192, miR-30a, and miR-16 have been reported to be non-coding RNAs closely associated with kidney injury. Increasing rno-miR-192 level was validated in the plasma of IRI rats compared to that of a sham rat group. Interestingly, the plasma hsa-miR-129 level increased in AKI patients compared to that of non-AKI patients (AUC-ROC = 0.673, 95%CI: 0.540–0.806) [24]. Plasma miR-30a levels in contrast-induced AKI (CI-AKI) rat model (Fold change: 1.71) and clinical CI-AKI patients (Fold change: 2.58) were altered, indicating a potential application for the diagnosis of CI-AKI ($AUC_{miR-30a-5p} = 0.802$, 95%CI: 0.730–0.874) [25]. The test results in clinical samples demonstrated that the level of urine hsa-miR-16 was significantly higher in critically ill patients with AKI (Fold change: 100). However, these results also showed that mmu-miR-16 levels increased within 2 h after hypoxia/reoxygenization treatment and subsequently decreased in in vitro cell experiments [26].

Other miR-30 family members with anti-fibrosis properties are observed upon dysregulation of miR-30a-3p which may affect angiogenesis and trophoblast proliferation in the placenta. hsa-miR-30a-3p regulates expression of fms-related tyrosine kinase 1 (FLT1) to induce preeclampsia, leading to serious adverse effects on pregnant women [27]. However, in a clinical study, when compared to participants with AKI stage 3 renal non-recovery, hsa-miR-30a-3p was decreased in the serum of participants with AKI stage 3 renal recovery (0.74 vs. 0.89), but this finding was not statistically significant [28]. Interestingly, hsa-miR-30a-3p expression was down-regulated in lung adenocarcinoma, which was largely involved in the regulation of Wnt and AKT signaling pathways [29]. miR-30a-3p has been reported to be dysregulated in clear renal cell carcinomas, serving as an effective biomarker for the early diagnosis of lung cancer [30]. mmu-miR-30c-2-3p is highly expressed in hypertonic mouse KC3AC1 cells and is involved in the regulation of mineralocorticoid receptor, suggesting a potential application of miR-30c-2-3p in renal insufficiency over the course of diabetic nephropathy [31].

The miR-200 family mainly inhibits the epithelial–mesenchymal transition (EMT) by targeting zinc finger E-box binding homeobox (ZEB). EMT has been recognized as playing pivotal roles in the process of renal fibrosis. In a study on the relationship between environmental renal toxicants and biomarkers of kidney injury, miR-200c was found to be highly expressed in subjects exposed to chromium [32]. miR-200a-3p is an oncogenic miRNA through regulation of the expression of deleted in liver cancer 1 (DLC-1) in gastric cancer [33].

miR-378 is a prognostic marker in cholangiocarcinoma and has increased levels in a cisplatin-induced rat kidney injury model [34]. miR-194 is abnormally up-regulated in the cell lines of pancreatic tubular adenocarcinoma and in tissues of esophageal cancer [35]. The targeting effect of miR-192 on the β1 subunit of the $Na^+/K^+$-ATPase results in poor survival among non-small cell lung cancer patients [36]. miR-16 has been reported to be down-regulated in the plasma of AKI patients suffering from kidney injury [37]. miR-15b levels are diminished in patients at the end stage of renal disease, and is related to the hemoglobin, phosphate level, and glomerular filtration rate [38].

The KEGG analysis revealed that potential target genes of the top 10 up- and down-regulated DE-miRNAs are enriched in cell function, cytoskeleton, inflammation, and oxidative stress-related signaling pathways (Figure 4). Axon guidance and actin cytoskeleton-

related pathways are involved in cytoskeleton rearrangements, and their disruption can lead to renal tubular cell injury and fibrosis [39]. Axon guidance plays a vital role in neuronal development and regeneration, regulating the cytoskeleton of growth cones and promoting growth cones to advance, pause, turn, or collapse [40]. In addition, axon guidance affects islet morphology and function by regulating the migration of endocrine progenitor cells, which impacts the level of blood glucose metabolism in the body [41]. Axon guidance molecules interact with receptors to affect the development, maintenance, and resolution of tumors and neuropathic pain through Wnt2 signaling, Eph-ephrin signaling, and the semaphorin–plexin system [42]. A study on perineural invasion of pancreatic ductal adenocarcinoma revealed the role of axon guidance molecules. Gene knockout technology was used to construct PLXND1 and Annexin A2 (ANXA2) knockout mouse models, which reduced innervation and metastasis. ANXA2 was found to promote perineuronal invasion and metastasis of pancreatic duct adenocarcinoma by promoting the secretion of the Axon Guidance molecule SEMA3D and binding to its receptor PLXND1 [43]. Actin-related proteins largely maintain the homeostatic structures of cells to support cell–cell and cell–matrix communication and interaction. AKI is characterized by a reduction in renal function, decreased glomerular filtration rate, and renal tubular necrosis. Cell necrosis and apoptosis may disrupt cell homeostasis and eventually lead to the reorganization of the actin cytoskeleton, resulting in abnormal cell–cell and cell–matrix interactions [44].

Phosphatidylinositol signaling [45] and the Hedgehog pathway [46] are involved in many cellular functions, such as cell proliferation, differentiation, and apoptosis. The insulin signaling pathway influences ROS levels through regulation of glucose homeostasis and insulin sensitivity [47]. Phosphatidylinositol signaling impacts diverse cellular functions (cell proliferation, differentiation, and apoptosis) by regulating synthesis of phosphatidylinositol and its phosphorylated forms. PI3Kγ belongs to the class I group of the phosphoinositide 3-kinase (PI3K) family. It was reported that PI3Kγ knockout attenuated kidney injury in an I/R-induced kidney injury mouse model. The PI3Kγ knockout mice had reduced levels of infiltrating neutrophils, macrophages, and T cells in the kidney samples and lowered levels of inflammation [48]. PI3Kγ could also mediate the inflow of macrophages, T cells, and fibroblasts into the kidney, leading to renal fibrosis and injury [49]. The Hedgehog (Hh) pathway plays a vital role in tissue formation, cell growth and differentiation, and maintenance of stem cell function. Sonic hedgehog (Shh) is one of the three ligands of Hh. The level of Shh was increased in injured renal tubular epithelial cells in an I/R-induced mouse model of renal fibrosis [50]. Olive leaf extract concentrated in hydroxytyrosol was able to inhibit I/R-induced renal cell apoptosis and improve renal function in rats. Hydroxytyrosol activates the Hh pathway through promotion of Shh expression [51]. As a lymphangiogenesis factor, Shh can mediate proliferation and expansion of lymphatic endothelial cells, and even participate in the regulation of renal fibrosis and lymphangiogenesis [52].

The insulin signaling pathway often involves regulation of insulin metabolism and the maintenance of intracellular glucose levels. Insulin disorders may lead to a series of adverse consequences such as inflammation. The nuclear factor-κB pathway is necessary to respond to the increase in oxytocin receptors during inflammation [53].

The potential target genes of DE-miRNAs were analyzed for their interactions using a PPI network, and hub genes were identified based on their connectivity degree using Cytoscape software. The hub genes (LYN, ERBB, MAPK1, RAC1, and KRAS) are related to the action signaling pathway, while the hub genes (GATA (3/4), TRPS1, IGF1R, RAP2C/KRAS) are involved in insulin signaling. Most hub genes were mainly related to cell structure and inflammatory response (Table 5). The analysis showed that *GATA4* and *MAPK1* were the central genes with the highest degree of connectivity. As a member of the GATA family, *GATA4* is reported to be required in cancer [54], and it may also play a role in AKI. *GATA4* can affect the progression of liver fibrosis by regulating the expression levels of fibrogenic and anti-fibrotic genes in hepatic stellate cells [55], and it can also co-operate with other signaling pathways, such as

Notch signaling and PI3K/AKT signaling, as a potential therapeutic target for tumor diseases [56]. As another member of the GATA protein family, *GATA3* has been validated to be expressed in many tumors (breast cancer, renal tumor, and endocrine tumor), and can be used as a diagnostic or prognostic indicator of tumor diseases [57]. *GATA3* is also associated with impaired adipogenesis and insulin resistance. Inhibiting the function of *GATA3* can promote the healthy distribution of adipose tissue, improve insulin sensitivity, and effectively reduce the risk of type 2 diabetes [58]. *MAPK1*, also known as *ERK2*, patriciates various cellular processes through the MAPK/ERK signaling pathway, including division, differentiation, and development [59]. mmu-miR-483-5p affects the process of renal interstitial fibrosis in type 1 and type 2 diabetic mice through regulation of the expression of *TIMP2* and *MAPK1* genes [60].

cfa-miR-452, cfa-miR-30a, cfa-miR-106a, and cfa-miR-106b regulated the highest number of hub genes (Figure 5). hsa-miR-452 is a tumor oncogene across several cancers and a diagnostic biomarker for bladder cancer [61]. In a septic AKI study, the Boston University mouse proximal tubular cell line was treated using lipopolysaccha­rides (LPS) in vitro, and a sepsis model was established by injection or cecal ligation. The results demonstrated that mmu-miR-452 expression was increased both in vivo and in vitro. Moreover, changes in mmu-miR-452 expression in serum and urine preceded renal dysfunction or tissue injury. The researchers compared 47 sepsis pa­tients with AKI, 50 patients without AKI, and 10 healthy subjects. The serum and urine hsa-miR-452 levels in sepsis patients with AKI were significantly higher than control patients. Notably, urinary hsa-miR-452 levels may reflect the risk of AKI in sepsis patients [62]. miR-30a is responsible for the epithelial–mesenchymal transition (EMT) in podocytes, and its level decreased in contrast-induced AKI in both rats and humans [63]. Expression of miR-106a was elevated in most malignant tumors, except renal carcinoma, glioma, and astrocytoma [64]. Down-regulation of miR-106a is associated with the general prognosis of the disease. hsa-miR-106a is down-regulated in the plasma of focal segmental glomerulosclerosis (FSGS) patients and is also related to FSGS remission [65]. hsa-miR-106b has been confirmed to serve a function as a prognostic biomarker for patients suffering from hepatocellular carcinoma receiving transcatheter arterial chemoembolization (TACE) [66].

In conclusion, we established a novel gentamicin-induced AKI model through ad­ministration of 40 mg/kg gentamicin intravenously to beagle dogs for 12 consecutive days. Pathological analysis indicated that the degree of renal tissue injury ranged from minimal to mild. In the early stage of AKI, there was no significant alteration in traditional serum markers between the two groups. Interestingly, our study provides the expres­sion profile of urinary miRNAs and identifies potential target genes in a canine model of gentamycin-induced AKI by RNA sequencing. cfa-miR-30a-3p and cfa-miR-106b are the most up- and down-regulated DE-miRNAs, respectively. cfa-miR-452, cfa-miR-30a, cfa-miR-106a, and cfa-miR-106b are presumed to be important functional miRNAs in the regulation of gentamycin-induced nephrotoxicity. The present study is likely to supply valuable information for a more thorough understanding of the functions of miRNAs as novel biomarkers for nephrotoxicity.

**Author Contributions:** Writing—original draft, B.S.; Funding acquisition, B.S. and L.C.; Data cu­ration, Y.-W.Y., Z.Q. and Y.-F.M.; Formal analysis, L.C. and R.-L.W.; Supervision, X.-B.Z.; Con­ceptualizaion, B.L.; Writing—review and editing, X.-B.Z. and B.L. All authors have read and agreed to the published version of the manuscript.

**Funding:** This work was supported by Key Science and Technology Programs of Beijing Poly­technic (2020Z035-KXZ), Scientific Research Project of Beijing Yicheng Cooperative Development Foundation in 2021-Public welfare projects of rare disease related topics (YJXJ-JZ-2021-0014) and the Construction of High-level Teachers in Beijing Municipal Colleges and Universities (2019)-Youth talent (CIT & TCD 201904047).

**Institutional Review Board Statement:** The animal study protocol was approved by the Institutional Animal Care and Use Committee (IACUC) of the National Center for Safety Evaluation of Drugs (NCSED) (protocol code: IACUC-2019-K003 and approval date: 16 April 2019).

**Informed Consent Statement:** Not applicable.

**Data Availability Statement:** Publicly available datasets were analyzed in this study. This data can be found here: https://www.ncbi.nlm.nih.gov/sra/PRJNA911945, (accessed on 14 December 2022).

**Acknowledgments:** The authors thank Li Sun and Gui-Lin Cheng for animal experiment assistance.

**Conflicts of Interest:** The authors declare no conflict of interest.

## Abbreviations

| Abbreviation | Latin |
| --- | --- |
| cfa | Canis familiaris |
| hsa | Homo sapiens |
| mmu | Mus musculus |
| rno | Rattus norvegicus |

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
