# Peer review of "Screening Differential Expression Profiles of Urinary microRNAs in a Gentamycin-Induced Acute Kidney Injury Canine Model"

_kidneydial, doi:10.3390/kidneydial3020019_

Round 1

Reviewer 1 Report

The study was to determine urinary microRNAs as an early marker for acute kidney injury in a beagle dog model of Gentamycin-induced AKI. The authors used the high-throughput sequencing technology to measure urinary microRNA levels and identified 234 DE-microRNAs. They further analyzed the top 10 upregulated- and downregulated microRNAs and their target genes which were mainly involved in the pathways associated with kidney injury. The authors concluded that miR-452, miR-106a, and miR-106b are involved in regulating a number of hub genes. The study addresses important matters regarding the prognosis or early detection of AKI, however, there are some major issues that need to be paid attention to.

Major concerns and suggestions

  1. The introduction section does not articulate well and needs to rewrite. Since it has been many studies focusing on detecting miRNAs as early markers of kidney injury in different AKI models, it would be better to focus on how the current study filled the gaps or improve our knowledge relevant to kidney injury rather than cancers.
  2. Need more information in the Methodology section to describe what criteria/scores for kidney histopathological grade of lessons were made.
  1. Since the circadian clock affects kidney function, what's the rationale to select this time period (16:30-8:30) for urine collection? Need detail on urine volume used for extraction of total RNA?
  2. What is the urinary miRNA profile in the control groups? It is unclear why this result was not included or served as the baseline instead of using the results from Day 3 post-gentamycin injection. Could the authors provide an explanation?
  3.   What are the actual fold changes on the top 10 upregulated and down-regulated DE-miRNAs? Is the P <0.05 an adjusted P valve?
  4.  Although Figure 4 shows GO annotation and KEGG pathway analysis, it would be easier for readers if a table was made for the text (lines 159-186) that is too long.
  5. Lines 325- 326, The most hub genes …. should it be referred to Table 3 rather than Table 2?
  6. Some descriptions are not reflected in what was found in the results section. For example, in Line 120, “We found that the dogs’ pathological features were not changed on day 12 in the control group”, should the pathological features were not found in the control group?
  7. A lot of statements in the introduction and discussion are lacking citations (e.g. Line 72, Lines 36-37, Line). Furthermore, some references cited are incorrect including the authors’ own publication (e.g. Lines 47-49, reference 9 does not match the statement). Line 267, reference 29 was a study in cholangiocarcinoma but not in a cisplatin-induced rat kidney injury model.
  8. The discussion section needs to be concise and focused more on the current findings related to kidney injury and relevant references. For example, the authors spent a lot of discussion on miRNA in a variety of cancers which is irrelevant to the current study.
  9. There are many grammar errors in the manuscript and most of them can be avoided by careful proofreading. For example, 3th and Line 25 in the Abstract; Lines 35 and  44-45 in the Introduction; even with others including some incomplete sentences or repeated sentences (Lines 335-337). 

Minor concerns

1.    Be consistent in using abbreviations such as DE-miRNAs, DE-microRNAs, or miR-. 

2. Some abbreviations need to be defined. For example, uts, mmu, hsa, or rno- need to be defined for readers who are not familiar with the field.

3. The statement in discussion (Line 211) is unclear the meaning of the mechanism of kidney injury is relatively simple” is.

4. Line 216, “The expression of sCr and BUN”, should it be better to say “the levels of sCr and BUN concentration? 

5. What’s the action proteins (Line 292) meant?

6. Line 193, degree= 60 but 58 in Table 3.

Author Response

We are very grateful to your comments for the manuscript. According with your advice, we amended the relevant part in manuscript. Some of your questions were answered below.

Q1. The introduction section does not articulate well and needs to rewrite. Since it has been many studies focusing on detecting miRNAs as early markers of kidney injury in different AKI models, it would be better to focus on how the current study filled the gaps or improve our knowledge relevant to kidney injury rather than cancers.

Response: The introduction section has been rewritten. Related studies on miRNAs as biomarkers of kidney injury have been added (line 47-55).

Q2. Need more information in the Methodology section to describe what criteria/scores for kidney histopathological grade of lessons were made.

Response: The criteria/scores for kidney histopathological grade of lessons has been added (line 83-95).

Q3. Since the circadian clock affects kidney function, what's the rationale to select this time period (16:30-8:30) for urine collection? Need detail on urine volume used for extraction of total RNA?

Response: Fasting urine sample was collected overnight in our study. Considering the maneuverability of the animals and the preservation of urine at 4℃, 16 hours (16:30-8:30) is the appropriate collection time . This method is widely used in the study of renal toxicity biomarkers (PMID: 30090426). 10-15 mL of urine samples were used for extraction of total RNA.

Q4. What is the urinary miRNA profile in the control groups? It is unclear why this result was not included or served as the baseline instead of using the results from Day 3 post-gentamycin injection. Could the authors provide an explanation?

Response: The main objective of this study was to investigate the trend of miRNA expression in urine at different treatment times (3, 6, and 12 days). Furthermore, we did not collect urine samples from the control group on days 6 and 12. It is not appropriate to use only day 3 urine results from the control group as a baseline.

Q5. What are the actual fold changes on the top 10 upregulated and down-regulated DE-miRNAs? Is the P <0.05 an adjusted P valve?

Response: The fold changes of the top 10 DE-miRNAs were added in table 2. P<0.05 is the adjusted P valve.

Q6. Although Figure 4 shows GO annotation and KEGG pathway analysis, it would be easier for readers if a table was made for the text (lines 159-186) that is too long.

Response: GO annotation and KEGG pathway were organized as table 3 and table 4.

Q7. Some descriptions are not reflected in what was found in the results section. For example, in Line 120, “We found that the dogs’ pathological features were not changed on day 12 in the control group”, should the pathological features were not found in the control group?

Response: According to the pathological tissue scoring criteria, pathological features were unchanged on day 12 in the control group. It have found that no pathological injury was observed in the kidney tissue of animals in the control group or even the low-dose group (PMID: 20118187/ 27074385).

Q8. A lot of statements in the introduction and discussion are lacking citations (e.g. Line 72, Lines 36-37, Line). Furthermore, some references cited are incorrect including the authors’ own publication (e.g. Lines 47-49, reference 9 does not match the statement). Line 267, reference 29 was a study in cholangiocarcinoma but not in a cisplatin-induced rat kidney injury model.

Response: References have been added and revised.

Q9. The discussion section needs to be concise and focused more on the current findings related to kidney injury and relevant references. For example, the authors spent a lot of discussion on miRNA in a variety of cancers which is irrelevant to the current study.

Response: The discussion section has been modified. Some of the DE-miRNAs screened in our study have not been reported in kidney injury studies. We used tumor-related reports to demonstrate the potential value of miRNAs in diagnosis of AKI.

Q10.There are many grammar errors in the manuscript and most of them can be avoided bycareful proofreading. For example, “3th” and Line 25 in the Abstract; Lines 35 and 44-45 in the Introduction; even with others including some incomplete sentences or repeated sentences (Lines 335-337). 

Response: The grammar errors and sentences have been modified.

Q11. Be consistent in using abbreviations such as DE-miRNAs, DE-microRNAs, or miR-. 

Response: The abbreviations used in the manuscript have been revised.

Q12. Some abbreviations need to be defined. For example, uts, mmu, hsa, or rno- need to be defined for readers who are not familiar with the field.

Response: Abbreviations have been added as supplementary (line 412).

Q13. The statement in discussion (Line 211) is unclear the meaning of “the mechanism of kidney injury is relatively simple” is.

Response: This sentence has been revised to “Although there are numerous drugs that can potentially induce nephrotoxicity in clinical practice, the mechanism of kidney injury is relatively straightforward” (line 226).

Q14. What’s the action proteins (Line 292) meant?

Response: “action proteins” was revised to “actin-related proteins” (line 309).

Reviewer 2 Report

- In the abstract, it may seem that you have done experiments in only one dog. Please, remove " model" and write that the experiments were done in 5 mongrel dogs.

- Urine was stored on ice after collecting it. But I think that the urine should have been collected in cold recipients. This may have altered the results.

- You are studying an AKI model, but you do not have an AKI as your results clearly show. Both creatinine and BUN are completely normal. I am not satisfied by your comments in the discussion about the inavility of these markers to accurately diagnose AKi, but  recent evidence suggests that even relatively mild injury or impairment of kidney function is manifested by small changes in serum creatinine (sCr) and/or urine output (UO). See https://www.ncbi.nlm.nih.gov/pmc/articles/PMC5198510/

Author Response

We are very grateful to your comments for the manuscript. According with your advice, we amended the relevant part in manuscript. Some of your questions were answered below.

Q1. In the abstract, it may seem that you have done experiments in only one dog. Please, remove " model" and write that the experiments were done in 5 mongrel dogs.

Response: The relevant sentence has been modified.

Q2. Urine was stored on ice after collecting it. But I think that the urine should have been collected in cold recipients. This may have altered the results.

Response: Urine samples were collected on ice. After collection, samples were centrifuged and stored at -70℃. The relevant sentence has been modified (line 97).

Q3. You are studying an AKI model, but you do not have an AKI as your results clearly show. Both creatinine and BUN are completely normal. I am not satisfied by your comments in the discussion about the inavility of these markers to accurately diagnose AKi, but  recent evidence suggests that even relatively mild injury or impairment of kidney function is manifested by small changes in serum creatinine (sCr) and/or urine output (UO). See https://www.ncbi.nlm.nih.gov/pmc/articles/PMC5198510/

Response: The low sensitivity and specificity of traditional serum biomarkers (creatinine and BUN) have been widely demonstrated (PMID: 33556265). The changes in serum creatinine and urine output do not reflect tubular function or injury. Gentamicin used in our study mainly absorbed in the proximal tubules, which can cause tubular cell toxicity and eventually induce AKI. The AKI models were confirmed by pathological testing, which is the gold standard for AKI diagnosis.

It is reasonable that serum biomarkers (creatinine and BUN) did not change significantly within 12 days of treatment.

Round 2

Reviewer 1 Report

Some minor errors need to be corrected.

1. Line 97: What was the urine collection time period? 16 hours (16:30-8:30) or 2 hours between 16:30-18:30 as mentioned in the revised manuscript? 

2. Lines 230-231: KIM-1 and NGAL should be defined in the Introduction section.

3. Line 312: ".....kidney injury mice model" should be "kidney injury mouse model". 

4. Line 344: Inhibiting the function of "what" can promote ......

Author Response

We are very grateful to your comments for the manuscript. According with your advice, we amended the relevant part in manuscript. Some of your questions were answered below.

  1. Line 97: What was the urine collection time period? 16 hours (16:30-8:30) or 2 hours between 16:30-18:30 as mentioned in the revised manuscript? 

Response: 16-h urine samples were collected in our study during 16:30-8:30. Related sentence has been revised.

  1. Lines 230-231: KIM-1 and NGAL should be defined in the Introduction section.

Response: KIM-1 and NGAL have been defined in the introduction section.

  1. Line 312: ".....kidney injury mice model" should be "kidney injury mouse model". 

Response: Related sentence has been revised.

  1. Line 344: Inhibiting the function of "what" can promote ......

Response: Inhibiting the function of GATA3 can promote the healthy distribution of adipose tissue, improve insulin sensitivity, and effectively reduce the risk of type 2 diabetes. Related sentence has been revised.
